# Radiological Assessment in Idiopathic Pulmonary Fibrosis (IPF) Patients According to MUC5B Polymorphism

**DOI:** 10.3390/ijms232415890

**Published:** 2022-12-14

**Authors:** Elisabetta Cocconcelli, Nicol Bernardinello, Chiara Giraudo, Gioele Castelli, Clorinda Greco, Roberta Polverosi, Marina Saetta, Paolo Spagnolo, Elisabetta Balestro

**Affiliations:** 1Respiratory Disease Unit, Department of Cardiac Thoracic Vascular Sciences, Public Health University of Padova, 35128 Padova, Italy; 2Department of Medicine-DIMED, Padova University Hospital, 35128 Padova, Italy; 3San Giovanni di Dio Hospital, 88900 Crotone, Italy; 4Antoniano Diagnostic Institute, 35123 Padova, Italy

**Keywords:** MUC5B genotype, HRCT scores, IPF survival

## Abstract

The *MUC5B* rs35705950 mutant T allele is the strongest genetic risk factor for familial and sporadic IPF. We sought to determine whether *MUC5B* genotype influences radiological patterns of IPF at diagnosis, as well as their change over time, in patients on antifibrotic therapy. Among eighty-eight IPF patients, previously genotyped for MUC5B rs35705950, we considered seventy-eight patients who were evaluated for radiological quantification of the following features both at treatment initiation (HRCT1) and after 1 year (HRCT2): ground glass opacities (AS), reticulations (IS) and honeycombing (HC). Of the evaluated patients, 69% carried at least one copy of the T allele (TT/TG). Carriers of the T allele displayed similar FVC loss in the first year of treatment as GG carriers, but overall survival at the end of follow-up was longer in the TT/TG group, compared to the GG group. In the GG group, both the AS and HC increased significantly, whereas in the TT/TG group only HC increased over the first year of treatment. *MUC5B* rs35705950 GG carriers are associated with increased ground glass and honeycombing extent over time and worse survival than T allele carriers. Longitudinal HRCT may help define the prognostic role of the *MUC5B* rs35705950 genotype.

## 1. Introduction

Idiopathic pulmonary fibrosis (IPF) is a chronic disorder of unknown origin, and the overall survival of affected patients is very poor and unpredictable [1]. The approved antifibrotic drugs pirfenidone and nintedanib are only able to slow down the inexorable disease progression; therefore, IPF remains a deadly disease, and the development of more efficacious and better tolerated therapies is an urgent priority.

Despite uncertainty about its leading cause, a number of potential risk factors have been suggested. Indeed, IPF is believed to occur in genetically susceptible individuals as a consequence of an aberrant wound-healing response following repetitive alveolar microinjury, resulting in scarring of the lung parenchyma and irreversible loss of function [2]. Familial clustering of cases and the occurrence of pulmonary fibrosis in the context of rare genetic disorders (such as Hermansky–Pudlak syndrome or dyskeratosis congenita) indicate that genetic predisposition contributes significantly to the pathogenesis of IPF [3].

The promoter polymorphism, rs35705950, within the mucin 5B (MUC5B) gene has been associated with both sporadic IPF and familial pulmonary fibrosis in several independent cohorts [4,5,6,7]. The increased risk of developing pulmonary fibrosis is conferred by carriage of the mutant allele (T), either in heterozygous (GT) or in homozygous (TT) form. The prognostic role of rs35705950 in IPF is still debated. Indeed, several authors did not observe any association between the MUC5B polymorphism and survival in different real-world cohorts, whereas in other studies, even considering different ethnicities, a protective effect on mortality for the minor T allele, both in homozygous and heterozygous forms, was shown [8,9,10,11,12]. We recently reported on the impact of *MUC5B* rs35705950 on the prognosis of IPF patients on antifibrotic treatment [13]. Specifically, we showed that carriers of the mutant T allele, either in heterozygous or homozygous form, displayed a significantly longer survival compared to patients homozygous for the wild type allele.

With this background, the aim of this study was to determine whether *MUC5B* rs35705950 genotype affects radiological patterns of IPF at diagnosis and whether, and to what extent, it influences disease progression, as assessed by HRCT, during the first year of antifibrotic treatment.

## 2. Results

### 2.1. Clinical and Functional Evaluation at Baseline and during the First Year of Follow Up

Baseline clinical and functional characteristics of the study population are shown in Appendix A. Of 88 patients genotyped for MUC5B rs35705950, HRCT scan at the first year of follow-up was not available for ten patients, who were excluded from the study. Based on the *MUC5B* rs35705950 genotype, 54 patients were classified as TT/TG and 24 as GG (Appendix A).

Twenty-seven (35%) patients died during the follow up period, with an equal proportion between the two *MUC5B* genotype groups. The allele frequency of the MUC5B rs35705950 alleles were in Hardy–Weinberg equilibrium, as previously described [66/156 (42%) for the minor T allele and 90/156 (58%) for the wild type G allele] (Appendix A) [13].

### 2.2. Radiological Score at Baseline

Alveolar, honeycombing, interstitial and pooled honeycombing and interstitial score in the HRCT performed at diagnosis (HRCT1) were similar between the two genotype groups (Appendix A). In particular, at baseline, AS was 22% (0–62) in TT/TG and 16% (0–44) in GG (*p* = 0.52), HC was 2% (0–41) in TT/TG and 3% (0–70) in GG (*p* = 0.54), IS was 22% (0–52) in TT/TG and 25% (0–45) in GG (*p* = 0.91), and HC + IS was 28% (9–73) % in TT/TG and 30% (8–89) in GG (*p* = 0.76) (Appendix A). Similarly, no difference in radiological scores at baseline were observed when TT and TG were considered separately (Appendix A).

The inter-observer agreement between the two radiologists regarding change in AS, IS and HC was good (Cohen’s kappa = 0.71 for IS, k = 0.76 for AS, k = 0.80 for HC).

### 2.3. Radiological Scoring during 1-st Year Follow Up

In the study population as a whole, HC and HC + IS increased significantly between HRCT1 and HRCT2 from 2% (0–70) to 6% (0–70, *p* < 0.0001) and from 28% (8–89) to 33% (8–98, *p* < 0.0001), respectively (Figure 1, Table 1). Conversely, AS and IS remained similar between HRCT1 and HRCT2 from 20% (0–62) to 20% (0–64, *p* = 0.16) and from 22% (0–52) to 23% (0–59, *p* = 0.12), respectively (Figure 1, Table 1).

When the study population was stratified by the MUC5B rs35705950 genotype, in TT/TG patients HC increased significantly between HRCT1 and HRCT2 from 2% (0–41) to 5% (0–63, *p* = 0.001) (Figure 2B), whereas AS and IS did not (from 22% (0–62) to 21% (0–64, *p* = 0.81) and from 22% (0–52) to 23% (0–59, *p* = 0.47) respectively; Figure 2A,C). Conversely, among GG patients, both AS and HC increased significantly from 16% (0–44) to 18% (1–86, *p* = 0.05) and from 3% (0–70) to 7% (0–83, *p* = 0.007), whereas IS remained similar between HRCT1 and HRCT2 (from 26% (0–45) to 26% (0–53, *p* = 0.15), respectively; Figure 2A–C).

When HC and IS were pooled together, the HC + IN score increased significantly in TT/TG patients (from 28% (9–73) to 30% (9–93), *p* = 0.001), and in GG patients (from 28% (8–89) to 42% (8–89), *p* = 0.002), respectively (Figure 2D).

When TT and TG were considered separately, HC and HC + IN increased only in the TG group and not in the TT group (Appendix A).

### 2.4. Survival Analysis and Multivariate Analysis

The overall survival of IPF patients carrying the TT/TG genotypes was longer than that of carriers of the GG genotype. Indeed, median survival was 69 months for TT/TG patients and 41 months for GG patients (HR 0.52, 95% CI 0.28–0.98; *p* = 0.04) (Figure 3).

Survival analysis of TT/TG and GG genotype patients. The black line represents survival in the TT/TG group and the blue line represents survival in the GG group. Kaplan Meier analysis was used with a log-rank test (HR 0.52, 95% CI 0.28–0.98; *p* = 0.04).

To assess whether radiological scores may be predictive of mortality in our IPF population, we used Cox proportional hazards regression analysis. Univariate analysis of radiological features associated with mortality revealed that IS and HC + IS in HRCT1, AS, IS and HC + IS in HRCT2, and the absolute increase in honeycombing were significantly associated with mortality in the entire IPF population (Table 2). On multivariate analysis, HC + IS in HRCT2 (HR: 1.02; 95% CI: 1.00–1.03; *p* = 0.01) remained an independent predictor of mortality in IPF patients under antifibrotic treatment (Table 2).

## 3. Discussion

In this study, we investigated, for the first time, the association between *MUC5B* rs35705950 genotype and radiological features, as assessed by HRCT, in a well characterized IPF cohort, both at baseline and after antifibrotic treatment. Despite similar radiological scores between the two groups (TT/TG and GG group) at treatment initiation, we observed that, after treatment, the alveolar score was significantly increased in GG patients but not in the TT/TG group, while the HC score was significantly increased in both groups. Of interest, carriers of the GG genotype showed a heavier smoking history and a lower respiratory function at treatment initiation compared to TT/TG carriers. When evaluating our study population as a whole, we observed that, at treatment start, on univariate analysis, IS and HC were significantly associated with mortality, while, after treatment, AS, IS and HC + IS were significantly associated with mortality. This latter, HC + IS, remained the independent predictor of mortality on multivariate analysis. In addition, carriers of the mutant rs35705950 T allele displayed better survival than non-carriers, regardless of the extension of HRCT changes at baseline, which was similar in the two groups.

Previous studies had demonstrated that a common variant (rs35705950) in the promoter region of the mucin 5b (*MUC5B*) gene was significantly associated with susceptibility to familial and sporadic IPF, suggesting a potential role for the distal airways and mucus overproduction in the pathogenesis of pulmonary fibrosis [6,14]. Specifically, it was reported that MUC5B rs35705950 T-carrier status was associated with increased expression of MUC5B glycoprotein in both distal airways and honeycomb cysts in IPF lungs [15,16] Interestingly, IPF patients carrying the minor T allele, either in homozygous or heterozygous form, appeared to have a better outcome, compared to IPF patients who did not carry the T allele, although the mechanisms underlying this association remain to be elucidated [8,17].

In line with previous reports, we confirmed the association between the rs35705950 T allele and longer survival in patients with IPF. Importantly, we confirmed this finding over a longer observation period (median time 52 months), compared to previous studies, as well as in patients on antifibrotic treatment.

The mechanism through which *MUC5B* rs35705950 T confers a survival advantage in patients with IPF is unknown. In an attempt to elucidate it, we genotyped for *MUC5B* rs35705950 in our population of IPF patients, who were further characterized both radiologically, by semi-quantitative scoring, and clinically. At baseline, patients carrying the TT/TG genotypes, when compared to GG carriers, had a less pronounced smoking history, better preserved lung function and a more frequent need for histological confirmation of the diagnosis, probably due to the atypical radiological features at first presentation. Theoretically, TT/TG carriers might be more symptomatic by virtue of mucus overproduction, which may contribute to mucociliary clearance dysfunction [15], thus medical advice is sought at an earlier stage, compared with those homozygous for the G allele. Indeed, carriage of the mutant T allele leads to excess mucus, particularly at the bronchoalveolar junction [18], and impaired mucociliary function [18,19,20]. In turn, this may result in prolonged retention of inhaled irritants (air pollutants, cigarette smoke, or microorganisms, among others), and, over time, lung injury, aberrant repair and fibrosis [21]. At present, however, this can only be speculated upon.

When we looked at the HRCT features at baseline, we did not observe any differences in terms of alveolar, honeycombing, or interstitial scores between carriers and non-carriers of the mutant T allele, suggesting that our patient population was homogeneous with regard to extent and type of lung damage at diagnosis and treatment start. This finding might also suggest that the radiologic appearance was not influenced by the presence or absence of the minor T allele, even though, in our cohort, we found that patients in the TT/TG group were more likely to need a histological confirmation of their diagnosis (i.e., they did not have a “definite” usual interstitial pneumonia pattern of fibrosis). Indeed, within the TT/TG group, half of the patients (57%) underwent surgical lung biopsy, compared to 29% of those carrying the GG genotype. This is in contrast with a previous study, by Chung and colleagues, in which pulmonary fibrosis patients carrying the T allele were less likely to display a disease pattern inconsistent with UIP (20.3% vs. 30.5% of carriers of the G allele), and more likely to have either a probable or definite UIP pattern (43.8% vs. 32.6% of carriers of the G allele) [22]. However, they looked at a spectrum of ILD, both idiopathic and non-idiopathic, whereas we specifically selected patients with IPF, and this may account for the apparently conflicting results. Of interest, in a Chinese IPF cohort, the authors did not find any significant associations between MUC5B promoter rs35705950 and the extent of honeycombing, likely due to the racial differences [23].

Pirfenidone and nintedanib are approved for treatment of IPF worldwide, based on their ability to slow down functional decline and disease progression with an acceptable safety and tolerability profile [24,25]. The trajectory of lung function decline in IPF is not linear; thus, at present, there are no tools that can reliably predict disease progression in individual patients [26,27]. In a previous study, we investigated whether HRCT features correlated with functional decline in treatment naïve patients with IPF and found that, at diagnosis, individuals with a higher alveolar score experienced a faster functional decline, compared with patients with a lower alveolar score [28], suggesting that extensive alveolar changes in HRCT may herald a more aggressive disease and worst outcome. In this study of IPF patients, stratified by *MUC5B* rs35705950 genotype, we did not observe any associations between *MUC5B* genotype and HRCT features of disease at baseline; conversely, after 12 months of antifibrotic treatment, the alveolar score increased significantly in the GG, but not in the TT/TG, genotype group, suggesting that GG genotype may increase the risk of radiological progression. This observation confirmed previous data suggesting that carriers of the rs35705950 G allele were more likely to display ground glass opacity on CT than carriers of the G allele in patients with idiopathic interstitial pneumonia [22]. In addition, on univariate analysis, the alveolar score after treatment was significantly associated with the risk of mortality, suggesting that alveolar opacity and rs35705950 GG genotype might identify a disease subset at higher risk of progression.

IPF is the archetypal progressive fibrotic interstitial lung disease that responds poorly to antifibrotic therapy [1,29]. We previously described the type, extent and evolution of radiologic abnormalities in patients with IPF after one year of antifibrotic treatment. Specifically, we found that the extent of honeycombing increased both in patients experiencing functional decline and in those who remained functionally stable, suggesting that HRCT might capture subtle features of disease progression [30]. In the present study, we expanded on this by demonstrating that the increase in the extent of honeycombing was similar in IPF patients on antifibrotic treatment, irrespective of the carriage of the T allele. In order to analyze the real impact of the minor T allele on HC increase, we separated the TT/TG and observed that the TT group presented no changes in radiological scores over time. Conversely, carrying the minor allele T in heterozygosity and the wild type form of the gene was associated with honeycombing increase over time. Honeycombing on HRCT is a well-established independent predictor of mortality in patients with IPF [26,31], and assessment of its presence and evolution over time remains critically important in disease monitoring and prediction of outcome, regardless of antifibrotic therapy.

Our results should be interpreted in light of some limitations. First, this was a single-center and retrospective study. However, some of our findings were in line with those of previous larger studies. Second, there was no standardized algorithm for the visual scoring of HRCT features of IPF. We used a previously described semi-quantitative scale. Reassuringly, the level of agreement between the radiologists was good. To date, no study has reported on the CT features of patients with IPF stratified by *MUC5B* rs35705950 genotype and treated for at least one year with antifibrotic therapy. Thus, our results, by combining HCRT features of IPF and *MUC5B* genotype data, may help predict disease behavior, and risk of progression and worse outcome. At present, neither the *MUC5B* rs35705950 genotyping nor semi-quantitative analysis of CT analysis is routinely performed; therefore, larger studies, and more robust evidence, are needed before these tools can be used to aid disease prognostication in IPF.

## 4. Materials and Methods

### 4.1. Study Population and Study Design

In the context of the aim of our study, we considered a cohort of IPF patients referred to our center between April 2014 and September 2018 and who were enrolled in our previous work [13], as previously described (Appendix A). All patients were genotyped for MUC5B rs35705950 by PCR amplification and Sanger sequencing and categorized as TT/TG genotype and GG genotype (Appendix A). For sample processing, DNA extraction and Sanger sequencing techniques, see additional file 1 in our previous work [13].

The patients were followed clinically, functionally and radiologically for at least one year after initiation of anti-fibrotic treatment (either pirfenidone or nintedanib), as previously described (Appendix A). For each patient, clinical and functional data was collected at the time of treatment initiation and at regular time intervals (every four months) thereafter, while HRCT was regularly performed at treatment initiation and after 12 months. Patients for whom at least two HRCT were not available (i.e., HRCT both at diagnosis and after one year of treatment) were excluded.

The overall survival (OS) was calculated from the beginning of the treatment to death, transplant, or loss to follow-up, as previously described, with survival data updated and censored in June, 2022.

### 4.2. Radiological Scoring 

The HRCT scans available at treatment initiation (HRCT1) and at the 12-month follow-up (HRCT2) were scored by three thoracic radiologists (C.G.; R.P.; C.G.). The HRCTs were performed by a 64 slice Siemens Somatom Sensation (Siemens Healthcare, Erlangen, Germany), applying a slice thickness of ≤1.5 mm.

The radiologists were blind to clinical and functional data and timing of HRCT, and scored the HRCT1 and HRCT2 images independently using a semi-quantitative scale as previously described [30]. This represented a modification of the previously reported scoring systems [28,32]. Specifically, the following radiologic features were considered: ground glass opacities (GGO), (alveolar score, AS), reticulations (interstitial score, IS) and honeycombing (HC) (HC score). For each lung lobe, the two radiologists assessed the extent of AS, IS and HC, using a scale from 0–100, and estimated extent to the nearest 5%. After each individual lobe was scored, the results were expressed as the mean value of the five lobes in AS, IS and HC. Finally, the IS and HC were pooled (IS + HC) to quantify the extent of fibrotic abnormalities. The level of interobserver agreement was obtained for each patient as the mean of 5 lobes, and for each radiological abnormality (i.e., IS, AS and HC), and expressed as Cohen’s k value. Disagreement between radiologists was resolved by consensus. The association between radiological change and FVC decline was calculated as the change in AS (ΔAS/month), IS (ΔIS/month), HC (ΔHC/month), pooled IS and HC (ΔIS + HC/month) and the change in FVC milliliters (ml) per month (ΔFVC ml/month) and FVC% pred. per month (ΔFVC% pred./month) between HRCT1 and HRCT2.

### 4.3. Statistical Analysis

Categorical variables were described as absolute (n) and relative values (percentage, %), whereas continuous variables were described as median and range. To compare demographic data and baseline clinical characteristics between stable TT/TG and GG genotypes, Chi square test and Fisher’s exact test for categorical variables and Mann–Whitney U test for continuous variables were used, as appropriate.

The Wilcoxon signed rank test was performed to compare HRCT1 and HRCT2 for the grading scores of different variables (AS, IS, HC and IS + HC) in the study population as a whole, and for TT/TG and GG patients. Correlation coefficients between radiological and functional data were calculated using the nonparametric Spearman’s rank method. The level of inter-observer agreement between two expert radiologists was assessed by kappa statistic measure [33].

The overall survival was calculated from treatment initiation to death, or lung transplantation, with data censured in June, 2022. The cumulative survival rate was calculated using the Kaplan–Meier method, and the difference in survival times between the two groups (TT/TG and GG genotypes) was assessed with a log-rank test. Radiological scores were evaluated for their relationship with survival in a univariate analysis of Cox proportional hazards regression testing. Variables with an association statistically significant with overall survival at univariate analysis were included in a multivariate Cox proportional hazard regression test to identify factors independently associated with mortality.

All data were analyzed using SPSS Software version 25.0 (IBM Corp., Armonk, NY, USA) and figures were created with GraphPad Prism (version 8.3.1, GraphPad Software, La Jolla, CA, USA). *P*-values < 0.05 were considered statistically significant.

## 5. Conclusions

In IPF patients on antifibrotic treatment, carriage of the *MUC5B* rs35705950 T allele, is associated with increased honeycombing over time, whereas non-carriers of the mutant allele experience increase of both ground glass opacity and honeycombing, despite a similar functional decline. Further studies are needed to identify the role of MUC5B polymorphism in disease development and progression in IPF.

## Figures and Tables

**Figure 1 ijms-23-15890-f001:**
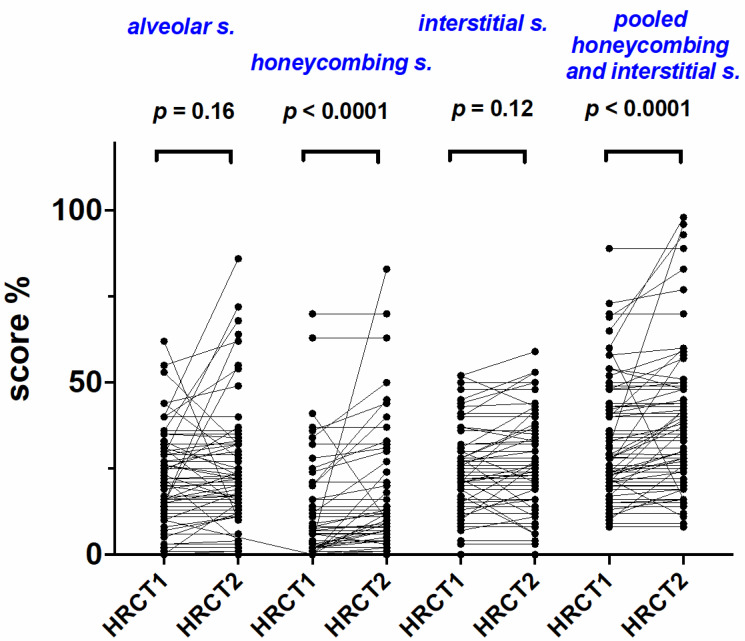
Radiological scores at treatment initiation (HRCT1) and after one year of treatment (HRCT2) in the entire study population. Change in alveolar score, interstitial score, honeycombing and pooled interstitial score and honeycombing at treatment initiation (HRCT1) and after one year of treatment (HRCT2) in the entire study population. *P* values refer to comparisons between HRCT1 and HRCT2 and Wilcoxon signed rank test for pared non parametric data was used.

**Figure 2 ijms-23-15890-f002:**
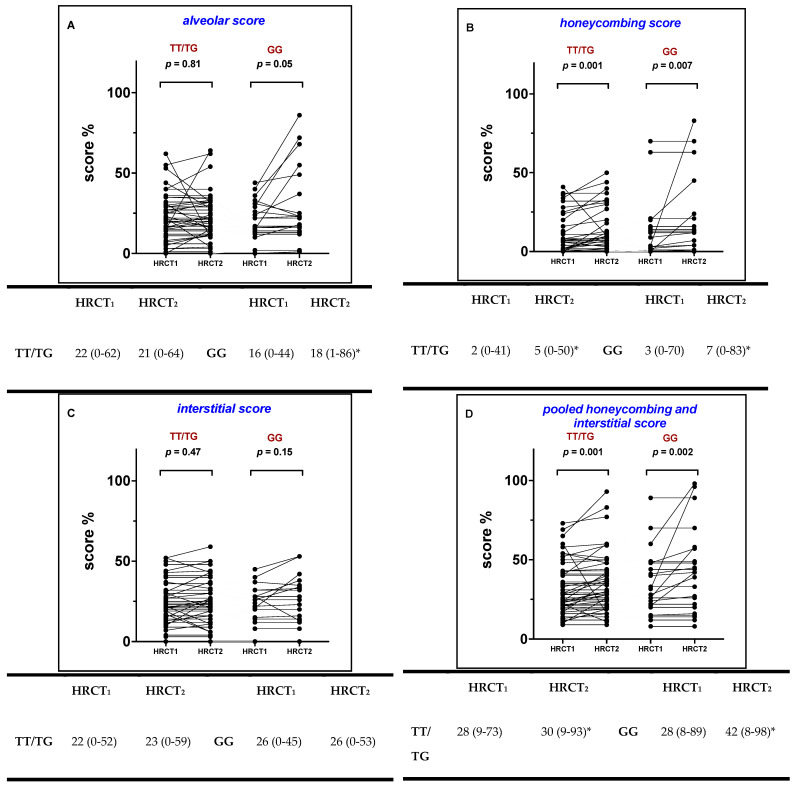
Change in alveolar score (**A**), honeycombing (**B**), interstitial score (**C**) and pooled interstitial score and honeycombing (**D**) at treatment initiation (HRCT1) and after one year of treatment (HRCT2) in TT/TG and GG genotype patients. The *p* values and * refer to comparisons between HRCT1 and HRCT2 and Wilcoxon signed rank test for paired non parametric data was used.

**Figure 3 ijms-23-15890-f003:**
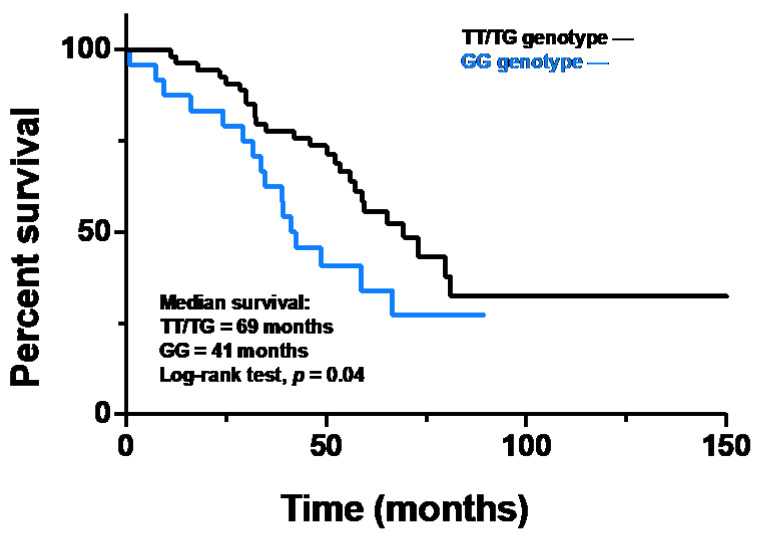
Survival of the study population categorized in TT/TG genotype or GG genotype.

**Table 1 ijms-23-15890-t001:** Radiological scores at treatment initiation (HRCT1) and after one year of treatment (HRCT2) in the entire study population.

	HRCT_1_	HRCT_2_	*p* Value		HRCT_1_	HRCT_2_	*p* Value
**AS**	20 (0–62)	20 (0–64)	0.16	**IS**	22 (0–52)	23 (0–59)	0.12
**HC**	2 (0–70)	6 (0–83)	**<0.0001**	**HC + IS**	28 (8–89)	33 (8–98)	**<0.0001**

AS = alveolar score; IS = interstitial score; HC = honeycombing; HC + IS = pooled interstitial score and honeycombing. Values are expressed as median and range. *P* values refer to comparisons between HRCT1 and HRCT2, and Wilcoxon signed rank test for paired non parametric data was used.

**Table 2 ijms-23-15890-t002:** Predictive factors of overall mortality in the entire population of IPF patients treated with antifibrotics.

	Univariate Analysis		Multivariate Analysis	
	HR (95% IC)	*p*	HR (95% IC)	*p*
**Alveolar score in HRCT1 (%)**	1.01 (0.99–1.03)	0.11	-	**-**
**Honeycombing in HRCT1 (%)**	1.00 (0.98–1.02)	0.52	-	-
**Interstitial score in HRCT1 (%)**	1.03 (1.00–1.05)	**0.01**	1.08 (0.99–1.17)	0.07
**Interstitial s. and honeycombing in HRCT1 (%)**	1.01 (1.00–1.03)	**0.02**	0.97 (0.93–1.01)	0.15
**Alveolar score in HRCT2 (%)**	1.02 (1.00–1.04)	**0.008**	1.01 (0.99–1.04)	0.15
**Honeycombing in HRCT2 (%)**	1.01 (0.99–1.03)	0.16	-	-
**Interstitial score in HRCT2 (%)**	1.03 (1.00–1.05)	**0.009**	0.94 (0.88–1.01)	0.14
**Interstitial s. and honeycombing in HRCT2 (%)**	1.02 (1.00–1.03)	**0.003**	1.02 (1.00–1.03)	**0.01**
**Change in Alveolar score (%)**	1.00 (0.99–1.05)	0.98	-	**-**
**Change in Interstitial score (%)**	1.02 (0.94–1.10)	0.44	-	**-**
**Change in Honeycombing (%)**	1.03 (1.01–1.06)	**0.03**	1.05 (0.97–1.13)	0.18
**Change in Interstitial s. and honeycombing (%)**	1.02 (0.99–1.05)	0.058	-	-

Values are expressed as HR (95% CI). Univariate and multivariate Cox proportional hazard regression tests were used to determine the relationship of radiological scores with mortality.

## Data Availability

Not applicable.

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
