# Peer review of "Radiological Assessment in Idiopathic Pulmonary Fibrosis (IPF) Patients According to MUC5B Polymorphism"

_ijms, 2022, doi:10.3390/ijms232415890_

Round 1
Reviewer 1 Report
Cocconcelli et al explore the impact of a Muc5B polymorphism on radiological assessments and patient outcomes in a cohort of IPF patients assessed at their Center between April 2014 and September 2018 via retroactive data analysis. They report as Results/Conclusions that: 1. 69% of patients carry the TT/TG allele, 2. The TT/TG group has a similar loss of FVC compared to the GG group; 3. Survival in the TT/TG group is higher than in the GG group; 4. Alveolar and honeycombing scores increased in the GG group, while only the honeycombing scores increased in the TT/TG group.
If all of these findings were novel, they would be of relevance to the field. However, what the authors present is a re-analysis of a patient cohort they have reported on in multiple manuscripts before with their major Results/Conclusions already reported previously/elsewhere (e.g. 1. 69% of patients carry the TT/TG allele, 2. The TT/TG group has a similar loss of FVC compared to the GG group; 3. Survival in the TT/TG group is higher than in the GG group; e.g. PMID: 33794872, 31540181, 30909411,…). Thus, data reported in Tables 1+2 and Figures 1 and 3 are not actually novel but have been shown in some form or other before (e.g. PMID: 33794872, 31540181, 30909411,…). As far as I can tell, the novelty and significance of the present manuscript are limited to Fig. 2A, which shows that AS scores are not significantly increased between HRCT1+2 in the TT/TG group, but happen to be p=0.05 in the GG group. Without any confirmation, validation, or further exploration, this finding by itself is of minor potential significance and impact. Should I be mistaken, and the current patient cohort is distinct from the patients the authors reported on before, the authors may be able to increase the significance of their study by assessing whether their findings/conclusions are confirmed/reproduced independently in the two cohorts of IPF patients.
Specific points:
- The authors should clearly indicate and reference whether this is a different cohort of patients compared to the IPF patients they reported on previously in multiple manuscripts (e.g. PMID: 33794872, 31540181, 30909411,…) or whether this is essentially the same (or largely overlapping) patient group. This is critical for several reasons: 1. If it is not clearly indicated that these are more or less the same patients, researchers performing meta-analyses using the published literature may inadvertently treat them as independent datasets, thus creating a bias towards this single cohort of patients; 2. If these are the same/overlapping patient cohorts, the authors should provide an explanation for the variation in the exact number of patients included in the various manuscripts; 3. The Materials and Methods section should be revised (even the Methods section of the Abstract); it presently leaves the distinct impression as if new hands-on experimentation was performed just for the present study (e.g. sample processing, DNA extraction, sequencing,…), which may not have been the case.
- It could be interesting to provide datasets separated into TT and TG genotypes (perhaps as supplemental data), rather than just the combined TT/TG group
- The authors should refer to the TT/TG group always in the same way, and not go back and forth between TT/TG (Ln58) and TT/GT (Ln59).
- Figure 3 needs labels/units for the x axis.
- Fig. 3: As there is no further death after about 70 months, it appears that patients (at least the TT/TG group) can be readily divided into non-survivors, that live no longer than 70 months, and survivors, that live to 150+ months. It may be interesting re-analyzing the radiological assessment data separating these two groups.
- The authors previously reported that higher alveolar scores at baseline predict worse outcomes, but do not reproduce this finding upon stratification of GG and TT/TG genotypes in the present study. Does the patient pool in its sum (GG+TT/TG) still produces this outcome?
-
Reviewer 2 Report
Manuscript ID: ijms-2023935
The article is distinguished by the originality of the selected topic, structured correctly, and written in Standard English. The manuscript presents the: IPF patients on antifibrotic treatment, carriage of the MUC5B rs35705950 T allele, 380 is associated with increased honeycombing over time, whereas non-carriers of the mutant 381 allele experience increase of both ground glass opacity and honeycombing, despite a sim382 ilar function decline.
The introduction, materials, methods, and results are presented correctly and a logical relationship between them is clearly observed. In an initial review of the article, there were stylistic and grammatical mistakes that I hope the authors will avoid after revision.
Suggest:
- Most of the literature used is from before 2018, and only 7 % (8 articles) are from the last 4 years. I suggest it be updated! I suggest adding at least 10 more literature sources to validate the results achieved.
- 2. The discussion part is well explained. I suggest being supported with materials from the last two 21-22 years. Some of the sentences are clumsy and with further editing, this will be avoided.
- I have no objections to the figures and tables and the graphic design!
Round 2
Reviewer 1 Report
The authors' revisions have sufficiently addressed my concerns regarding duplication and misrepresentation of the novelty of the data/study.
The revised study is still of low impact, given that novel insights are minimal compared to what the authors have already published.